# The Association between Cardiovascular Risk Factors and Lichen Sclerosus: A Systematic Review and Meta-Analysis

**DOI:** 10.3390/jcm13164668

**Published:** 2024-08-09

**Authors:** Suvijak Untaaveesup, Piyawat Kantagowit, Nattawut Leelakanok, Petcharpa Chansate, Wongsathorn Eiumtrakul, Walaiorn Pratchyapruit, Chutintorn Sriphrapradang

**Affiliations:** 1Phaholpolpayuhasena Hospital, Kanchanaburi 71000, Thailand; suvijak2541@gmail.com; 2Faculty of Medicine, Chulalongkorn University, Bangkok 10330, Thailand; kantpiya@windowslive.com; 3Division of Clinical Pharmacy, Faculty of Pharmaceutical Sciences, Burapha University, Chonburi 20131, Thailand; nattawut.le@go.buu.ac.th; 4Faculty of Medicine, Thammasat University, Pathum Thani 12120, Thailand; praewa.chansate@gmail.com; 5Department of Medicine, Faculty of Medicine Ramathibodi Hospital, Mahidol University, Bangkok 10400, Thailand; wongsathorn.eiu@outlook.com; 6Institute of Dermatology, Ministry of Public Health, Bangkok 10400, Thailand; itesatuk@gmail.com

**Keywords:** vulvar lichen sclerosus, diabetes mellitus, obesity, metabolic syndrome, hypertension, dyslipidemia

## Abstract

**Background/Objective**: Lichen sclerosus is a chronic inflammatory skin disease that affects people of all ages and sexes. Evidence of cardiovascular risk factors in lichen sclerosus has been continuously reported; however, the definitive association remains inconclusive. This meta-analysis aimed to summarize the association between cardiovascular risk factors and lichen sclerosus. **Methods**: Electronic databases, including MEDLINE and EMBASE, were systematically searched from inception to May 2024 to identify the literature reporting the association between cardiovascular risk factors and lichen sclerosus. A random-effects model was used for the meta-analysis. **Results**: We included 16 eligible studies: nine case–control studies, six retrospective cohort studies, and one cross-sectional study. A total of 432,457 participants were included. Lichen sclerosus was significantly associated with type 2 diabetes mellitus with an odds ratio of 2.07 (95% CI: 1.21–3.52). Although not statistically significant, a trend of increasing risk in hypertension, dyslipidemia, obesity, and metabolic syndrome was observed among lichen sclerosus patients, with odds ratios of 1.56 (95% CI: 0.90–2.70), 1.44 (95% CI: 0.94–2.23), 5.84 (95% CI: 0.37–92.27), and 1.36 (95% CI: 0.52–3.54), respectively. **Conclusions**: Lichen sclerosus was associated with diabetes mellitus and potentially correlated with hypertension, dyslipidemia, obesity, and metabolic syndrome. Population-based prospective observational studies are required to further elucidate these findings and assess the impact of these associations.

## 1. Introduction

Lichen sclerosus (LS) is a chronic, benign, inflammatory skin disease characterized by inflammation, epithelial thinning, and distinctive dermal changes. LS typically occurs in the genital area, predominantly in females, with two onset peaks: prepubertal and postmenopausal age groups. In young individuals, LS involves prepubertal children and can manifest as hypoestrogenism [1,2]. Lesions in extragenital areas are mostly asymptomatic, while those in genital areas can present as glittering, whitish, patchy, and hyperkeratotic lesions. Clinical symptoms or signs may include pruritus, pain, balanoposthitis, phimosis, scarring, and stricture in affected organs [3]. In male patients, the lesions can involve the penis glans, urethra, and foreskin. Symptoms include phimosis, urethral stricture, and paraphimosis [4,5]. In female patients with LS, lesions occur in the inner areas of the labia minora, labia majora, clitoral hood, and perianal area. This anogenital involvement can lead to stenosis of the introitus, adherence of the labia minora, phimosis of the clitoris, and difficulties with defecation [1].

Although the exact prevalence of LS remains unknown [6], it was estimated to be 3% to 13% of older adult females in specialist clinics [7,8] and was likely underreported since one-third of the cases were asymptomatic [8]. The incidence of LS is increasing, as can be seen by the age-standardized incidence rate among Dutch women, which increased from 5 per 100,000 person–years in 1997–1998 to 35.7 per 100,000 person–years in 2021–2022 [9]. The etiology of LS is not fully understood [3], but probable mechanisms include hereditary factors, an autoimmune phenotype, and genetic alterations. LS is associated with microvessel injury, characterized by endothelial necrosis and basement membrane thickening. The link between inflammatory change and microvascular injury remains unclear, although the presence of C5b-9 in vessels of LS patients suggests a potential role in vessel injury [10]. 

The association between LS and cardiovascular risk factors may be explained by the aforementioned mechanisms, similar to other chronic inflammatory skin diseases. For instance, inflammation and the autoimmune nature of LS may lead to diabetes mellitus, whereas microvascular damage from LS may lead to hypertension. Uncontrolled diabetes mellitus and hypertension can cause further microvascular injury [11,12]. Nonetheless, the association between LS and cardiovascular risk factors remains inconclusive. Previous relevant studies showed an increasing risk of cardiovascular risk factors in individuals with LS [13,14,15,16]. Gulin et al. showed a significant correlation between LS and the risk of type 1 diabetes mellitus (OR = 1.9; 95% CI 1.6–2.1) [17]. In addition, Hieta et al. demonstrated a significant risk of cardiovascular risk factors, including type 2 diabetes mellitus, hypertension, and dyslipidemia among LS patients [17]. However, certain studies demonstrated a decreased risk of these comorbidities in LS [18,19,20]. Furthermore, Liu et al. reported no bidirectional association between diabetes mellitus or body mass index (BMI) and LS [21]. Given the inconsistency in existing results, this systematic review and meta-analysis were performed to summarize the association between cardiovascular risk factors and influencing factors of the association to provide comprehensive evidence. 

## 2. Materials and Methods

This systematic review and meta-analysis adhered to the Preferred Reporting Items for Systematic Reviews and Meta-Analyses (PRISMA) 2020 statement (Appendix A) [22] and the Meta-analyses Of Observational Studies in Epidemiology (MOOSE) checklists (Appendix A) [23]. The protocol was registered before the initiation of this study (INPLASY202460045) [24]. 

### 2.1. Search Strategy

Four investigators (SU, PK, PC, and EC) independently constructed the search strategy to include relevant published articles. The search was conducted in the MEDLINE and EMBASE databases from inception to May 2024. The search strategy was based on terms related to LS and cardiovascular risk factors (diabetes mellitus, hypertension, dyslipidemia, obesity, and metabolic syndrome) (Appendix A). To ensure comprehensiveness, we also manually searched each reference of potentially included articles. 

### 2.2. Inclusion Criteria

We included eligible studies, including cross-sectional, cohort, or case–control studies, that reported the risk of diabetes mellitus, metabolic syndrome, hypertension, dyslipidemia, or obesity in patients with LS. These studies needed to use patients without LS as their controls. Eligible studies were limited to those published in the English language. 

The articles were screened in two rounds. The first round was for title and abstract screening, and the second was for full-text review. Four investigators (SU, PK, PC, and EC) independently screened the articles. Any discrepancies were resolved through discussions with the senior investigators (NL and CS).

### 2.3. Data Extraction

Four investigators (SU, PK, PC, and EC) independently extracted the following topics to the standardized data extraction form: (1) baseline characteristics, including the last name of the first author, publication year, country, type of study design, study period, number of total participants, locations of LS, diagnostic criteria for LS, percentage of males, mean age of participants, and mean BMI of participants; (2) outcomes of interest, including the risk of diabetes mellitus, metabolic syndrome, hypertension, dyslipidemia, or obesity in patients with LS compared with patients without LS, reported as an adjusted and unadjusted odd ratio (OR), relative risk (RR), or hazard risk ratio (HR), with their 95% confidence interval (CI); and (3) adjusted factors for statistical analysis. Corresponding authors of articles potentially included were contacted to retrieve missing information for comprehensiveness. 

### 2.4. Quality Assessment

Two researchers (SU and PK) used the Newcastle–Ottawa quality assessment scale for case–control, cohort, and cross-sectional studies to assess the quality of the included articles [25]. Any conflicts were resolved through consultation with NL. 

### 2.5. Risk of Bias Assessment

Three investigators (SU, PK, and PC) independently assessed the risk of bias in the included studies using the Quality In Prognosis Studies (QUIPS) tool. This tool includes six domains (study participants, study attrition, prognostic factor measurements, outcome measurement, confounding, and statistical analysis and reporting). The risk of bias was reported as high, moderate, and low. Any of the conflicts were resolved by consulting NL.

### 2.6. Certainty of Evidence

Three investigators (SU, PK, and PC) independently evaluated the certainty of evidence in the included studies using the Grading of Recommendations, Assessment, Development, and Evaluations (GRADE) tool. This tool includes five domains (risk of bias, inconsistency, indirectness, imprecision, and publication bias). The certainty was categorized as high, moderate, low, and very low. Any conflicts were resolved by consulting NL.

### 2.7. Statistical Analysis

The association between cardiovascular risk factors and LS was meta-analyzed using Review Manager 5.3 software (clicktime.com, Inc., San Francisco, CA, USA) from the Cochrane Collaboration. A random-effect model was used due to variations in study methodologies and settings. Overall effect sizes were estimated using the Mantel–Haenszel method. Heterogeneity among included studies was examined using the I^2^ statistic. A value of I^2^ between 0 and 25% indicated insignificant heterogeneity, 51–75% represented moderate heterogeneity, and I^2^ greater than 75% signified high heterogeneity [26]. Meta-regression by age, sex, study design, quality of studies, study country region, and the use of pathology for diagnosis was planned a priori. Heterogeneity was further examined through sensitivity analyses by removing one study with different methodologies at a time. Publication bias was assessed through visual inspection of the funnel plot without performing a statistical test for asymmetry. Significant asymmetry indicated potential publication bias or heterogeneity [27]. If studies reported both numbers for patients with type 1 and type 2 diabetes mellitus, only those with type 2 were included in the analysis.

## 3. Results

### 3.1. Search Results

The systematic search retrieved 1123 potential articles (791 from EMBASE and 332 from MEDLINE), of which 247 were duplicates and removed. Subsequently, the titles and abstracts of 876 articles were screened. After reviewing 81 full-text articles, 65 were excluded due to their irrelevant outcomes, unrelated study designs, or the unavailability of articles in English. As a result, 16 articles were included in the analysis (Figure 1) [13,14,15,16,17,18,19,20,28,29,30,31,32,33,34,35].

### 3.2. Baseline Characteristics of Eligibility Studies

A total of 432,457 participants from 16 articles were included. The included studies comprised nine case–control studies [15,16,17,18,19,29,30,31,32], six retrospective cohort studies [13,14,20,28,33,34], and one cross-sectional study [35]. Of these, eight studies [14,15,19,20,28,31,33,34] were from the Americas, seven [13,16,17,18,29,32,35] from Europe, and one [30] from the Eastern Mediterranean Region. Table 1 demonstrates the baseline characteristics with quality assessment of the eligibility articles. The diagnostic criteria for LS in each study are detailed in Appendix A.

### 3.3. Risk of Bias Assessment

The risk of bias assessment classified four studies [14,29,31,33] as low risk and twelve studies [13,16,17,18,19,20,28,30,32,33,34,35] as moderate risk (Appendix A). The overall risk of bias assessment in each study is demonstrated in Appendix A. A high risk of bias assessments was found in only bias due to confounding domain. Appendix A depicts the summarized proportion of the risk of bias assessment for each domain.

### 3.4. Certainty of Evidence Assessment

Appendix A describes the certainty of evidence assessment from the Grading of Recommendations, Assessment, Development, and Evaluations (GRADE) tool. Evidence supporting the outcomes of diabetes mellitus and hypertension was graded as low certainty. The other comorbidities were supported by very low certainty of evidence.

### 3.5. Associated Outcomes in Lichen Sclerosus Patients 

#### 3.5.1. Diabetes Mellitus

Thirteen studies with 18,192 LS patients were included in the meta-analysis of the risk of diabetes mellitus [13,14,15,16,17,18,19,20,29,30,32,34,35]. The pooled unadjusted odds ratio (OR) was 2.07 (95% CI: 1.21–3.52, I^2^ = 96%), which was statistically significant (*p* = 0.008, Figure 2). Meyrick Thomas et al. [35] were excluded from the analysis due to the absence of event numbers. Of the included studies, only Halonen et al. [18] reported a decreased risk of diabetes mellitus. 

#### 3.5.2. Hypertension

Nine studies [14,15,17,18,19,20,28,30,33] (13,412 LS individuals) reported their unadjusted OR with hypertension outcome (Figure 3). The pooled OR was 1.56 (95% CI: 0.90–2.70, I^2^ = 98%), which was not statistically significant (*p* = 0.12). Of these, six [14,17,20,28,30,33] reported a significantly increased risk of hypertension, whereas Halonen et al. [18] reported a decreased risk.

#### 3.5.3. Dyslipidemia

Five studies [14,15,16,17,18] with 11,398 LS subjects were included in the analysis (Appendix A). Of these, only one [18] indicated a decreased risk of dyslipidemia in LS patients, while three [14,16,17] noted an increased risk of dyslipidemia among LS patients. The meta-analysis showed that LS was not associated with the risk of dyslipidemia (pooled unadjusted OR = 1.44, 95% CI: 0.94–2.23, I^2^ = 85%, *p* = 0.10).

#### 3.5.4. Obesity

The outcomes regarding obesity in 11,255 LS participants were investigated in three studies [18,28,31], as depicted in Appendix A. The meta-analysis showed that LS did not significantly increase the risk of obesity (pooled unadjusted OR = 5.84, 95% CI: 0.37–92.27, I^2^ = 89%, *p* = 0.21). Blaschko et al. [28] and Fuchs et al. [31] reported an enhanced risk of obesity in LS patients, while Halonen et al. [18] presented an insignificant risk of obesity. 

#### 3.5.5. Metabolic Syndrome

Two studies [18,20] revealed the OR of metabolic syndrome in LS, as shown in Appendix A. The pooled unadjusted OR was 1.36 (95% CI: 0.52–3.54, I^2^ = 92%), without statistical significance (*p* = 0.53). Only one [20] showed a positive association between metabolic syndrome and LS. 

### 3.6. Publication Bias

Appendix A provides an overall biases assessment for each outcome. The publication bias in this systematic review was investigated using the funnel plot from data on diabetes mellitus in LS patients since diabetes mellitus was the only outcome with sufficient publications (Appendix A). The funnel plot was not symmetrical; therefore, publication bias was probable. 

### 3.7. Meta-Regression

Because of the high heterogeneity, meta-regression was conducted. In diabetes mellitus studies, the study location (*p* = 0.003) and the study quality score (*p* = 0.009) were identified as the causes of heterogeneity. In hypertension studies, the study location, as classified by the World Health Organization (WHO) (*p* = 0.020), percentage of male participants (*p* = 0.030), average age of participants (*p* = 0.016), and study design (*p* = 0.024) were causes of heterogeneity. Table 2 summarizes the meta-regression analysis based on the baseline characteristics of eligible studies. We did not perform meta-regress for other outcomes due to the limited number of included articles.

### 3.8. Sensitivity Analysis

In the sensitivity analyses, by removing Halonen et al. [18], significant ORs with I^2^ reduction in diabetes mellitus and hypertension in LS were demonstrated as follows: 2.04 (95% CI: 1.68–2.48, I^2^ = 31%, *p* < 0.001) and 1.78 (95% CI: 1.35–2.35, I^2^ = 83%, *p* < 0.001), respectively (Figure 4 and Figure 5). 

## 4. Discussion

This meta-analysis demonstrated a significant association between diabetes mellitus and LS, with a 2.04-fold increasing risk, along with observed rising trends for hypertension, dyslipidemia, obesity, and metabolic syndrome in patients with LS. Sensitivity analyses revealed a 2.04-fold and 1.78-fold significantly elevated risk for diabetes mellitus and hypertension in LS patients, respectively. Halonen et al. [18] was the only study showing a significant decrease in the risk of diabetes mellitus and hypertension in our meta-analysis of LS patients. 

Our results depicted the significantly enhanced risk of diabetes mellitus in LS patients, which can be explained by inflammatory processes and microvascular diseases. The pathogenesis of LS is superficial vessel damage, which represents necrosis of the endothelium or microvessel walls and a decrease in size and density in dermal papillae under microscopic findings. The C5b-9 mediators play a significant role in endothelial injury due to their deposition in superficial vessels. The concurrence of diabetes mellitus in LS is linked to microvascular impairment through vascular dropout, which diminishes vessel density through C5b-9 mediators in some studies [10,11,36]. Furthermore, the deregulatory of inflammatory molecules, including cytokines and adipocytokines, may be involved in the development of diabetes mellitus in LS patients [37].

This study revealed a significant association between hypertension and LS. Hypertensive diseases cause microvascular injury and endothelial dysfunction through glycosylation of the other molecules of C5b-9 (CD59). C5b-9 accumulates in endothelial tissues, which causes the basement membrane layer enlargement. The other mechanism is to diminish the capillary density (rarefaction), which is caused by the resistance of vessels. However, the pathogenesis of rarefaction has not been elucidated [10,11,12,38,39]. 

The pathogenesis of LS is the genetic alteration that involves reactive oxygen species (ROS), causing changes in macromolecules (DNA, proteins, and lipids). The alteration leads to conditions such as cancer, autoimmune, and sclerosis. A previous study revealed a correlation between oxidative stress and LS pathogenesis [40]. Genetic alteration is also linked to oxidative stress in LS. High activation and production of wild-type p53 can induce oxidation reactions, which can lead to the development of cancerous lesions. Conversely, loss of p53 and genetic alteration of CDKN2A can precipitate cancer [41]. These ROS and oxidative stress could be linked to elevated cardiovascular risk factors in our study.

Increased cardiovascular risk factors observed in other chronic inflammatory skin diseases in adults, such as psoriasis [42,43] and atopic dermatitis [44,45,46], are similar in LS. While the specific inflammatory pathways differed among these conditions, these diseases shared a common aspect of chronic inflammation [42,47,48], which possibly led to increased risks. Nonetheless, evidence elucidating the mechanisms by which these comorbidities exacerbated LS or precipitated cardiovascular events remains limited in LS.

Management of cardiovascular risk factors in LS has not been emphasized. Recent guidelines mentioned an observed link between diabetes mellitus and BMI in LS patients [49,50]. Despite this, only obese male patients with LS and a buried penis are recommended to lose weight through multidisciplinary approaches [49]. In contrast, multidisciplinary approaches to all cardiovascular risk factors in patients with psoriasis are widely suggested [51,52]. Given the potential for elevated cardiovascular risk factors, healthcare providers should consider adopting a multidisciplinary approach for monitoring and managing cardiovascular health in selected patients with LS, similar to strategies for psoriasis. In addition, prompt and frequent evaluation and monitoring of cardiovascular risk factors may benefit certain selected LS patients [53]. Current diabetes guidelines emphasize reducing cardiovascular and renal complications and recommend glucose-lowering drugs with cardiorenal benefits. With the increasing use of sodium–glucose cotransporter receptor-2 (SGLT2) inhibitors, clinicians should be aware that these medications can cause vulvar pruritus, rash, and candidiasis, which may be mistaken for LS. It is important for clinicians to differentiate between these conditions to ensure the appropriate treatment for each [54,55].

Our findings robustly highlight the necessity for vigilant cardiovascular screening and monitoring in LS patients. However, the study was limited by the retrospective nature and the heterogeneity of the included studies, potential publication bias, and the lack of studies from certain geographic regions, including Africa, the Western Pacific, and Asia, which could affect the generalizability of the findings. The search terms may not include incorrect spelling, such as ‘lichen sclerosis’, and synonyms of LS, such as ‘balanitis xerotica obliterans’. However, we manually screened the keywords ‘lichen sclerosis’ and ‘balanitis xerotica obliterans’ to ensure comprehensiveness, resulting in no additional references. Prospective observational studies are needed to address these limitations and confirm our findings. 

Based on our findings, further studies are needed to establish more evidence on the risk of each type of diabetes mellitus and the interaction between sex, locations of involvement (genital/extragenital involvement), and cardiovascular risk factors in LS patients. Crucially, there is a lack of evidence supporting the correlation between LS and cardiovascular diseases such as cerebrovascular disease, acute coronary syndrome, and atherosclerosis. Therefore, future research should investigate the risk of cardiovascular diseases and mortality in LS patients.

## 5. Conclusions

LS is likely correlated with diabetes mellitus and possibly associated with hypertension, dyslipidemia, obesity, and metabolic syndrome. Population-based prospective observational studies are required to further elucidate the findings and assess the impact of these associations.

## Figures and Tables

**Figure 1 jcm-13-04668-f001:**
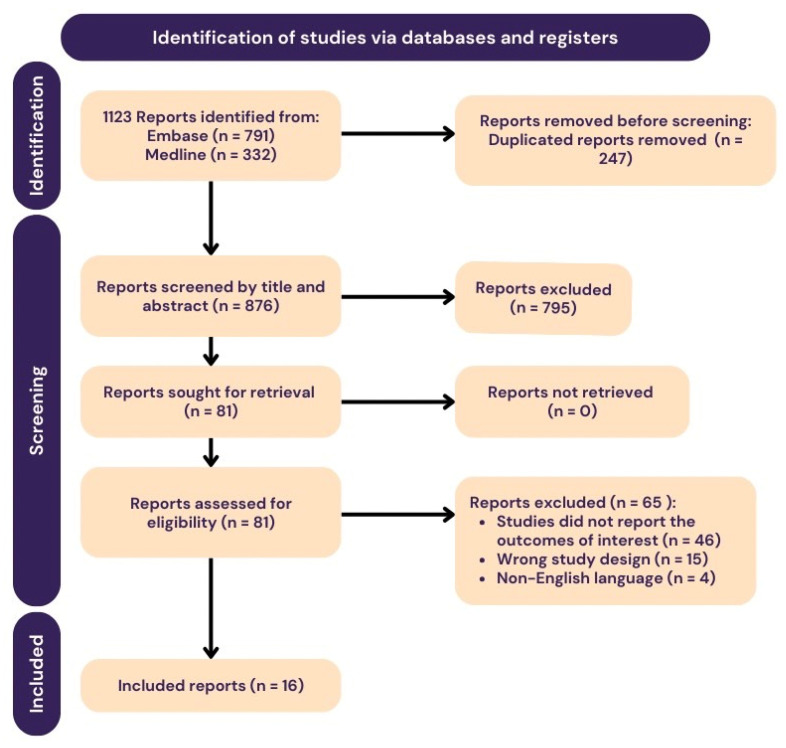
Flow diagram according to the PRISMA guideline.

**Figure 2 jcm-13-04668-f002:**
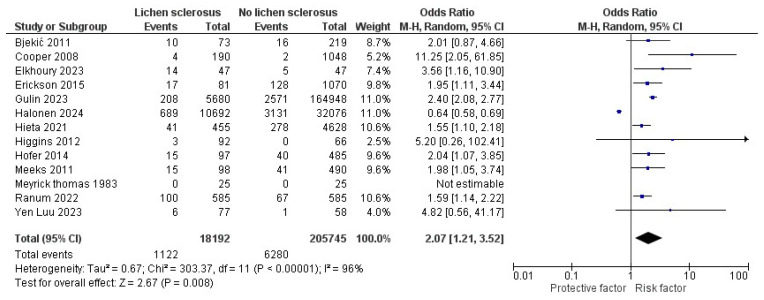
The forest plot for the pooled unadjusted odds ratio of diabetes mellitus in patients with lichen sclerosus [13,14,15,16,17,18,19,20,29,30,32,34,35].

**Figure 3 jcm-13-04668-f003:**
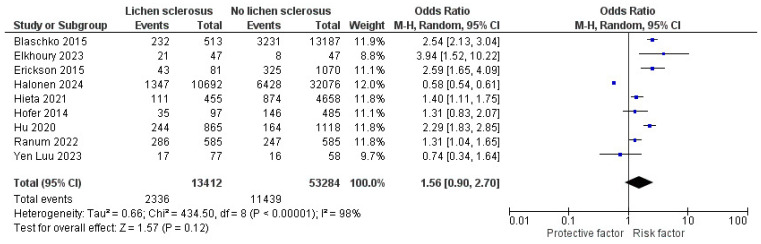
The forest plot for the unadjusted pooled odds ratio of hypertension in patients with lichen sclerosus [14,15,17,18,19,20,28,30,33].

**Figure 4 jcm-13-04668-f004:**
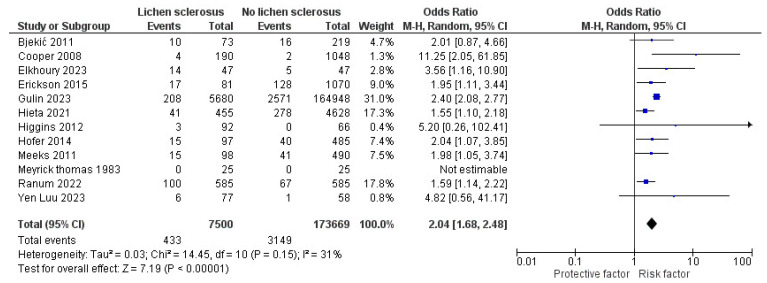
The sensitivity analysis of the forest plot for the pooled unadjusted odds ratio of diabetes mellitus in patients with lichen sclerosus [13,14,15,16,17,19,20,29,30,32,34,35].

**Figure 5 jcm-13-04668-f005:**
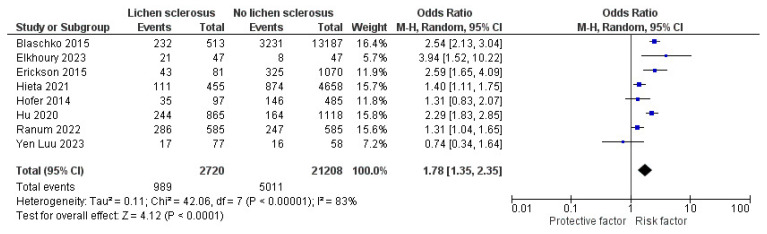
The sensitivity analysis of the forest plot for the pooled unadjusted odds ratio of hypertension in patients with lichen sclerosus [14,15,17,19,20,28,30,33].

**Table 1 jcm-13-04668-t001:** The baseline characteristics and quality assessment of the eligible studies.

Reference	Country	Study Design	Study Period/Duration	Total Participants, n	Male Sex, n (%)	Mean Age, Years (SD)	Mean BMI (SD)	Location of LS	Confounders	Newcastle-Ottawa Scale
Bjekić 2011 [16]	Serbia	CC	2007 to 2008	292	292 (100)	All participants < 25 years: 26 (8.9%), 25–34 years: 54 (18.5%), 35–44 years: 40 (13.7%), 45–54 years: 48 (16.4%), 55+ years: 124 (42.5%) *	NR	Genital area	NA	Selection: 3, Comparabilit-y: 1, Outcome: 3
Blaschko 2015 [28]	US	RC	2000 to 2010	13,700	13,700 (100)	LS group: 50.4 non-LS group: 45.3	NR	Urethra, specifically related to urethral stricture disease	age	Selection: 4, Comparabilit-y: 0, Outcome: 2
Cooper 2008 [29]	UK	CC	2 years	1238	0 (0)	LS group: 63 non-LS group: 62	NR	Vulva	NA	Selection: 3, Comparabilit-y: 1, Outcome: 2
Elkhoury 2023 [30]	Lebanon	CC	2010 to 2020	94	94 (100)	LS group: 49.8 (22.9) non-LS group: 45.7 (20.3)	LS group: 26.00 (2.72) non-LS group: 25.55 (3.57)	Genital area	NA	Selection: 2, Comparabilit-y: 1, Outcome: 3
Erickson 2015 [14]	US	RC	NR	1151	1151 (100)	LS group: 51.3 (13.8) non-LS group: 45.7 (16.6)	LS group: 34.6 (7.9) non-LS group: 29.6 (6.7)	Urethra, specifically related to urethral stricture disease	age	Selection: 3, Comparabilit-y: 1, Outcome: 3
Fuchs 2017 [31]	US	CC	2008 to 2015	300	300 (100)	LS group: 10.3 non-LS group: 10.1	LS group: 70.7 non-LS group: 52.4	Penile skin	NA	Selection: 4, Comparabilit-y: 0, Outcome: 3
Gulin 2023 [13]	Sweden	RC	2001 to 2021	368,248	LS: 1802 (31.7) non-LS: 184,848 (50.9)	LS group: 57 ^+^ non-LS group: 40 ^+^	NR	Anogenital region	age, sex	Selection: 3, Comparabilit-y: 0, Outcome: 3
Halonen 2024 [18]	Finland	CC	1998 to 2016	42,768	0 (0)	LS group: 60.8 non-LS group: 60.8	NR	Vulva	NA	Selection: 4, Comparabilit-y: 1, Outcome: 3
Hieta 2021 [17]	Finland	CC	2004 to 2012	NA	0 (0)	LS group: 64.4 non-LS group: NR	NR	Genital and perianal areas	NA	Selection: 3, Comparabilit-y: 1, Outcome: 2
Higgins 2012 [32]	UK	CC	NR	158	0 (0)	LS group: 65.0 (58.5, 71.5) ^+^ non-LS group: 46.0 (37.5, 53.0) ^+^	NR	Anogenital region	NA	Selection: 4, Comparabilit-y: 1, Outcome: 2
Hofer 2014 [15]	US	CC	1986 to 2009	582	582 (100)	LS group: 46.6 non-LS group: 48.3	LS group: 31.0 non-LS group: 28.1	Urethra, penis, prepuce, and scrotum	NA	Selection: 3, Comparabilit-y: 1, Outcome: 2
Hu 2020 [33]	US	RC	1996 to 2019	1983	0 (0)	LS group: 54.4 (15.7) non-LS group: 42.9 (16.0)	NR	Vulva	NA	Selection: 3, Comparabilit-y: 2, Outcome: 3
Meeks 2011 [34]	US	RC	1986 to 2009	588	588 (100)	NR, but LS group age was matched within 3 years with non-LS group.	LS group: 31.0 non-LS group: 28.1	Urethra, penile skin, prepuce, and scrotum	NA	Selection: 3, Comparabilit-y: 2, Outcome: 3
Meyrick thomas 1983 [35]	UK	CS	NR	50	50 (100)	LS group: 43 non-LS group: 44	NR	Penis, trunk, and limbs	NA	Selection: 3, Comparabilit-y: 1, Outcome: 3
Ranum 2022 [20]	US	RC	2015 to 2019	1170	0 (0)	Total participants: 57.3 *	LS group: 29.8 (7.1) non-LS group: 29.3 (7.6)	Genital and body areas	NA	Selection: 3, Comparabilit-y: 1, Outcome: 3
Yen Luu 2023 [19]	US	CC	2019 to 2021	135	0 (0)	LS group: 71 non-LS group: 54	LS group: 29.8 (8.4) non-LS group: 24.9 (5.5)	Vulva	NA	Selection: 4, Comparabilit-y: 1, Outcome: 3

Abbreviations: BMI, body mass index; CC, case–control study; CS, cross-sectional study; LS, lichen sclerosus; NA, not available; NR, not reported; RC, retrospective cohort study; SD, standard deviation; UK, United Kingdom; US, United States. * The data are presented in the form available from the study. ^+^ median (IQR) is shown instead of mean (SD).

**Table 2 jcm-13-04668-t002:** Meta-regression for the association between lichen sclerosus and the risk of diabetes mellitus and hypertension.

Possible Source of Heterogeneity	Number of Included Studies	Coefficient [95% CI]	*p*-Value
**Diabetes Mellitus**			
Male (%)	13	0.004 [−0.002 to 0.011]	0.165
Age (mean)	13	−0.012 [−0.075 to 0.052]	0.718
Study design	13		0.866
Case–control study	8	0.532 [0.098 to 0.966]	0.016
Retrospective cohort study	4	0.148 [−0.494 to 0.790]	0.651
Cross-sectional study	1	−0.532 [−4.602 to 3.538]	0.798
Quality of study by NOS stars	13	−0.399 [−0.699 to −0.099]	0.009
Study country	13		0.003
Serbia	1	0.700 [−0.251 to 1.651]	0.149
England	2	1.324 [−0.547 to 3.195]	0.166
Finland	2	−0.800 [−1.815 to 0.215]	0.122
Lebanon	1	0.571 [−0.964 to 2.105]	0.466
Scotland	1	0.949 [−2.211 to 4.109]	0.556
Sweden	1	0.532 [0.098 to 0.966]	0.745
United States	5	−0.074 [−1.084 to 0.936]	0.886
WHO regions	13		0.515
EUR	7	0.467 [0.038 to 0.897]	0.033
AMR	5	0.193 [−0.450 to 0.836]	0.556
EMR	1	0.803 [−0.645 to 2.251]	0.277
Pathology for diagnosis			0.289
Yes	8	−0.427 [−1.217 to 0.362]	0.289
No	2	1.036 [0.283 to 1.788]	0.007
**Hypertension**			
Male (%)	9	0.007 [0.001 to 0.013]	0.030
Age (mean)	9	−0.052 [−0.095 to −0.010]	0.016
Study design	9		0.024
Case–control study	5	−0.642 [−1.201 to −0.084]	0.024
Retrospective cohort study	4	0.737 [0.342 to 1.131]	<0.001
Quality of study by NOS stars	9	−0.308 [−0.685 to 0.069]	0.110
Study country	9		0.020
United States	6	0.567 [0.242 to 0.891]	<0.001
Finland	2	−0.696 [−1.296 to −0.097]	0.023
Lebanon	1	0.804 [−0.419 to 2.027]	0.198
WHO regions	9		0.020
EUR	2	−0.696 [−1.296 to −0.097]	0.023
AMR	6	0.567 [0.242 to 0.891]	<0.001
EMR	1	0.804 [−0.419 to 2.027]	0.198

Abbreviations: AMR, Region of the Americas; EMR, Eastern Mediterranean Region; EUR, European Region; NOS, Newcastle–Ottawa Scale; WHO, World Health Organization. The WHO regions’ classification was derived from the World Health Organization. (2023). World Health Statistics 2023 (ISBN-13: 9789240074323). World Health Organization.

## Data Availability

The original contributions presented in the study are included in the article/Appendix A; further inquiries can be directed to the corresponding authors.

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
