# Peer review of "The Association between Cardiovascular Risk Factors and Lichen Sclerosus: A Systematic Review and Meta-Analysis"

_jcm, 2024, doi:10.3390/jcm13164668_

Round 1
Reviewer 1 Report
Comments and Suggestions for Authors
This is a very well written meta analysis on the association between cardiovascular risk factors and lichen sclerosus. The methodology is precise, the results are clearly presented and possible clinical implications are discussed. I have only minor comments
- The updated S3 guideline for lichen sclerosus (2024) should be included (PMID 38822598)
- The authors may consider citing the following article by Liu et al 2024 (PMID 38689755), which showed no significant causal relationship between DM and BMI with LS in both directions
- Is there any evidence supporting correlation of LS with hard CV endpoints (e.g. stroke, acute myocardial infarction etc) or subclinical atherosclerosis?
Author Response
The revisions have been highlighted in blue in the manuscript.
- The updated S3 guideline for lichen sclerosus (2024) should be included (PMID 38822598)
Responses: We have added the reference number 52 in the line 297.
Revision (Discussion, page 11, lines 297)
Recent guidelines mentioned an observed link between diabetes mellitus and BMI in LS patients. [51,52]
- The authors may consider citing the following article by Liu et al 2024 (PMID 38689755), which showed no significant causal relationship between DM and BMI with LS in both directions
Responses: We have added the explanation in the line 77-78, as described.
Revision (Introduction, page 2, lines 77-78)
Furthermore, Liu et al. reported no bidirectional association between diabetes mellitus or body mass index (BMI) and LS. [22]
- Is there any evidence supporting correlation of LS with hard CV endpoints (e.g. stroke, acute myocardial infarction etc) or subclinical atherosclerosis?
Responses: The systematic search retrieved no article indicating a correlation between hard CV endpoints and LS. We have also added this point to the discussion.
Revision (Discussion, page 11, lines 323-326)
Crucially, there is a lack of evidence supporting the correlation between LS and cardi-ovascular diseases such as cerebrovascular disease, acute coronary syndrome, and atherosclerosis. Therefore, future research should investigate the risk of cardiovascular diseases and mortality in LS patients.
Reviewer 2 Report
Comments and Suggestions for Authors
What was your search strategy? It is important to imclude the terms that you searched for. You need to include balanitis xerotica obliterans, lichen sclerosis (incorrect spelling) as you can often pick up more papers.
Do you have data about the difference between males and females and gential/extra genital disease
Page 2, line 49, LS does not affect the urethra in females. It does not affect/the vagina - it is the inner labia minora/majora and clitoral hood. It is confusing as written and would be better split into separate groups
Is there different data between types 1 and 2 diabetes
Comments on the Quality of English LanguageSatisfactory
Author Response
The revisions have been highlighted in green in the manuscript.
- What was your search strategy? It is important to imclude the terms that you searched for. You need to include balanitis xerotica obliterans, lichen sclerosis (incorrect spelling) as you can often pick up more papers.
Responses: Our search strategy is depicted in Table S3, as described on page 2, line 93, and edited the PRISMA diagram has been updated accordingly. We manually screened the keywords of lichen sclerosis and balanitis xerotica obliterans to ensure comprehensiveness due to potential misspellings, resulting in no additional reference. This is described in the line 315-319, page 11.
Revision (Discussion, page 11, lines 315-319)
The search terms may not include incorrect spelling, such as ‘lichen sclerosis’, and synonyms of LS, such as ‘balanitis xerotica obliterans’. However, we manually screened the keywords ‘lichen sclerosis’ and ‘balanitis xerotica obliterans’ to ensure comprehensiveness, resulting in no additional reference.
- Do you have data about the difference between males and females and gential/extra genital disease
Responses: To date, there is no data on the differences between sexes and area involvement in LS patients. We have also included this point to the discussion.
Revision (Discussion, page 11, lines 321-323)
Based on our findings, further studies are imperative to establish more evidence on the risk of …, area involvement, and the interaction between sex and cardiovascular risk factors in LS patients.
- Page 2, line 49, LS does not affect the urethra in females. It does not affect/the vagina - it is the inner labia minora/majora and clitoral hood. It is confusing as written and would be better split into separate groups
Responses: We have corrected the information in the line 49-54, page 2.
Original (Introduction, page 2, lines 49-51)
The involved areas are the urethra, vaginal introitus, perianal areas, and foreskin, leading to dyspareunia, erectile dysfunction, difficulty in defecation, urethral stricture, and anorgasmia.
Revision (Introduction, page 2, lines 49-54)
In male patients, the lesions can involve the glans penis, urethra, and foreskin. Symptoms include phimosis, urethral stricture, and paraphimosis. [4,5] In female patients with LS, lesions occur in the inner areas of the labia minora, labia majora, clitoral hood, and perianal area. This anogenital involvement can lead to stenosis of the introitus, adherence of the labia minora, phimosis of the clitoris, and difficulty with defecation. [1]
- Is there different data between types 1 and 2 diabetes
Responses: To date, there is no data on the differences between types 1 and 2 DM in LS patients. We have also included this point to the discussion.
Revision (Discussion, page 11, lines 321-323)
Based on our findings, further studies are imperative to establish more evidence on the risk of each type of DM, area involvement, and the interaction between sex and cardiovascular risk factors in LS patients.
- Moderate editing of English language required.
Responses: This revised version has been reviewed by a native English speaker.
Reviewer 3 Report
Comments and Suggestions for Authors
I read with great interest your systematic review and metanalysis regarding the Association between Cardiovascular Risk Factors and Lichen Sclerosus
Introduction is clear, providing general info regarding LS, for scientists familiar and non-familiar to it, and moreover highlight the scarcity of evidence and the contra-results of them (some studies suggest strong association with HTN, DM2 and dyslipidemia, while other not) regarding its association with cardiometabolic risk factors and the need for such metanalysis on the field. As mentioned by the authors, the presence of C5b-9 in vessels of LS patients prompts a common pathway between vascular injury and inflammation and LS pathophysiology.
The authors have clearly stated their search strategy, data extraction, quality assessment, ROB assessment and the analysis they’ve performed.
They demonstrated:
- Strong association with DM2 (odds ratio (OR) was 2.07)
- Non-significant association with HTN (opposite results of trials)
- No association with dyslipidemia
I was delighted that your performed meta-regression analysis due to opposite results and high heterogeneity, as well as the sensitivy analysis after removing Halonen trial
- OR for DM 2.04 (p < 0.001)
- OR for HTN 1.78 (p < 0.001)
The discussion clearly suggests potential pathophysiological links between LS and DM2.
You could add this ref : https://pubmed.ncbi.nlm.nih.gov/36129363/ and suggest the clinical tip of when treating CV patients with SGLT2, SGLT2-associated dermatological SAEs should not be misdiagnosed as LS and be treated with steroids.
I have no other comments and should be totally Accepted with the minor revision of
- the reference of each trial addition in Tables
- and the addition of this SGLT2 reference
I would like to congratulate you for this excellent work.
Author Response
The revisions have been highlighted in pink in the manuscript.
- You could add this ref : https://pubmed.ncbi.nlm.nih.gov/36129363/ and suggest the clinical tip of when treating CV patients with SGLT2, SGLT2-associated dermatological SAEs should not be misdiagnosed as LS and be treated with steroids.
I have no other comments and should be totally Accepted with the minor revision of
- the reference of each trial addition in Tables
- and the addition of this SGLT2 reference
I would like to congratulate you for this excellent work.
Responses: We have added the reference for each included study in table 1, and the SGLT 2 reference on page 11 lines 304-310.
Revision (Discussion, page 11, lines 304-310)
Current diabetes guidelines emphasize reducing cardiovascular and renal complications and recommend glucose-lowering drugs with cardiorenal benefits. With the increasing use of sodium-glucose cotransporter receptor-2 (SGLT2) inhibitors, clinicians should be aware that these medications can cause vulvar pruritus, rash, and candidiasis, which may be mistaken for LS. It is important for clinicians to differentiate between these conditions to ensure appropriate treatment. [56,57]
Round 2
Reviewer 2 Report
Comments and Suggestions for Authors
Page 2, line 48. You are describing clinical signs here and not symptoms.
It would be helpful if possible to group the papers into male/female, genital/extra-genital as this may impact results more.
Comments on the Quality of English LanguageNo issues
Author Response
Page 2, line 48. You are describing clinical signs here and not symptoms.
Response: We have edited that sentence according to your comments, changing "clinical symptoms" to "clinical symptoms or signs".
It would be helpful if possible to group the papers into male/female, genital/extra-genital as this may impact results more.
Response: We strongly agree that grouping by sex and location of involvement would enhance the results. However, some studies lack critical details, such as the location of genital or extragenital involvement and the number of participants by sex, which makes such grouping unfeasible. We have highlighted these limitations in the discussion (Page 11, Line 322-323).
Minor editing of English language required
Response: We have a native English speaker to help us thoroughly edit the manuscript.